# Current Status and Future Applications of Robotic Surgery in Upper Gastrointestinal Surgery: A Narrative Review

**DOI:** 10.3390/cancers17121933

**Published:** 2025-06-10

**Authors:** Koichi Okamoto, Takashi Miyata, Taigo Nagayama, Yuta Sannomiya, Akifumi Hashimoto, Hisashi Nishiki, Daisuke Kaida, Hideto Fujita, Shinichi Kinami, Hiroyuki Takamura

**Affiliations:** Department of General and Digestive Surgery, Kanazawa Medical University Hospital, 1-1 Daigaku, Uchinadamachi, Kahoku 920-0293, Ishikawa, Japan; t-miyata@kanazawa-med.ac.jp (T.M.); nagayama@kanazawa-med.ac.jp (T.N.); stttty@kanazawa-med.ac.jp (Y.S.); ah-idai@kanazawa-med.ac.jp (A.H.); hisashi@kanazawa-med.ac.jp (H.N.); kaida-d@kanazawa-med.ac.jp (D.K.); hfujita@kanazawa-med.ac.jp (H.F.); kinami@kanazawa-med.ac.jp (S.K.); takamuh@kanazawa-med.ac.jp (H.T.)

**Keywords:** esophageal cancer, gastric cancer, minimally invasive surgery, polysurgery, robotic surgery

## Abstract

In recent years, robot-assisted surgery has emerged as a promising technique for treating gastrointestinal malignancies. In Japan, laparoscopic gastrectomy is the standard treatment for early-stage gastric cancer, and robot-assisted surgery is particularly well-suited for such procedures. Moreover, the precision of robot-assisted surgery can improve both short- and long-term postoperative outcomes in advanced gastric cancer treatment. Additionally, robot-assisted surgery offers a minimally invasive approach with significant benefits for esophageal cancer surgery. This review examines the role of robot-assisted surgery in enhancing procedural and oncological outcomes for patients with upper gastrointestinal cancer and explores its applications in conversion cases involving borderline resectable gastric or esophageal cancer and polysurgery cases in which adhesions or tumor invasion restrict surgical procedures.

## 1. Introduction

Robot-assisted surgery (RAS) has gained increasing popularity in gastrointestinal cancer treatment [1,2,3,4,5,6,7]. Since prostate surgery became eligible for insurance coverage in 2012, RAS has rapidly expanded, and nearly all gastroenterological cancer surgeries are currently covered by insurance. By integrating evidence-based surgical techniques with robotic technology, RAS can reduce patient burden and improve treatment outcomes.

Minimally invasive surgery (MIS) using robotic systems began with the development of the DaVinci Surgical System (DVSS) by Intuitive Surgical in 1995. Designed to make open surgery less invasive while enhancing safety and precision, the DVSS has undergone multiple upgrades to improve its clinical applicability [8]. The most widely used model as of 2025, the da Vinci Xi^®^, features enhanced mobility, compact flex joints, and minimal space occupancy during surgery. Key advantages include: (a) high-resolution 3D magnified visualization with operator-controlled endoscopy; (b) stabilization of hand tremors and motion scaling; (c) increased internal range of motion, improved patient access, and minimized external collisions; and (d) a variety of forceps and devices tailored for surgical procedures.

The advantages of RAS for surgeons include (a) reduced physical fatigue [9], (b) fewer range-of-motion restrictions, (c) superior learning curves compared with laparoscopic surgery [10,11,12], (d) ergonomic benefits [13], and (e) enhanced visibility of anatomical structures. Leveraging these advantages enables surgeons to apply appropriate tension and countertraction to tissues, accurately delineate resection margins, and excise tumors with minimal residual tissue while ensuring patient safety.

RAS allows for the switching of ports for forceps and scope insertion, facilitating seamless role transitions between instruments without concerns related to mirror imaging or eye–hand coordination. This feature may reduce procedural difficulties and enhance maneuverability in the surgical field.

With advancements in cancer treatment and the increasing number of older patients and cancer survivors [14], it is essential to consider both initial gastrointestinal cancer surgeries and MIS for cases involving multiple prior surgeries, metachronous remnant gastric cancer [15,16,17,18,19,20,21,22,23,24,25,26,27,28,29,30], conversion surgery for initially unresectable tumors following multidisciplinary treatment, and salvage surgery for residual tumors after chemoradiotherapy [31,32]. To improve surgical outcomes in such complex cases, discussions on the application of MIS, including RAS, are necessary. However, the safety and oncological efficacy of MIS in these contexts remain contentious.

This review provides an overview of the role of RAS in upper gastrointestinal cancer surgery, focusing on oncological outcomes and reduced invasiveness.

## 2. Materials and Methods

This narrative review was conducted by searching PubMed for relevant articles using the keywords: “gastrointestinal cancer,” “esophageal cancer,” “gastric cancer,” “laparoscopic surgery,” “minimally invasive surgery,” “robotic surgery,” “robot-assisted surgery,” “remnant gastric cancer,” “conversion surgery,” and “salvage surgery.” Reference lists of the included studies and related commentary were manually screened for additional relevant studies. Only randomized controlled trials and cohort studies with adequate data on treatment and prognostic outcomes were included, whereas studies lacking such data were excluded.

## 3. Minimally Invasive Surgical Approaches for Upper Gastrointestinal Cancer

Upper gastrointestinal cancers, including esophageal cancer (EC) and gastric cancer (GC), are more prevalent in Asia than in Western countries. Surgery remains the primary curative treatment for both EC and GC. In Japan, the reported surgical mortality rate for EC is 3.2%, and despite the increasing adoption of minimally invasive esophagectomy (MIE), the complication rate remains high at 41.9% [33].

Additionally, Japanese clinical studies on extensive surgery for advanced GC have not demonstrated a survival benefit for procedures such as prophylactic splenectomy for upper GC [34], bursectomy [35], or prophylactic para-aortic nodal dissection [36]. Consequently, interest in the surgical treatment of advanced GC has shifted toward MIS.

To reduce the invasiveness of radical surgery for EC and GC, thoracoscopic and laparoscopic MIS techniques are increasingly utilized, with emerging evidence supporting their efficacy [37,38,39,40,41,42,43,44,45,46,47,48,49,50,51,52]. However, MIS for EC and GC presents challenges, including a relatively high incidence of recurrent laryngeal nerve paralysis and pancreatic fistula [42,49]. The potential of RAS to mitigate these issues has spurred ongoing clinical research [7,53,54,55,56,57].

Comparative studies have demonstrated the advantages of robotic gastrectomy (RG) over laparoscopic gastrectomy (LG) for GC [6,7,53,54,58,59,60,61,62,63,64,65,66,67,68,69,70,71,72,73,74], robotic colectomy over laparoscopic colectomy for colorectal cancer [4], and robot-assisted minimally invasive esophagectomy (RAMIE) over conventional MIE or open esophagectomy for EC [3,32,56,57,58,75,76,77,78,79,80,81,82].

### 3.1. RAS for EC

Table 1 presents studies that compared the efficacy of MIE and RAMIE with that of open esophagectomy [3,31,50,51,56,57,75,76,77,78,79,80,81]. RAMIE, a recognized MIS technique for treating EC, is typically performed with the patient in the prone position under bilateral lung ventilation. This approach has been shown to reduce the incidence of respiratory complications. However, some studies have reported on the feasibility of performing RAS with the patient in the left lateral decubitus position, which may offer an advantage in cases requiring emergency conversion to thoracotomy or in borderline resectable tumors, where conversion from the prone position is technically challenging [80,82].

Sun et al. [12] and van der Sluis et al. [56] reported that RAMIE outperformed conventional MIE and open esophagectomy in terms of reduced blood loss, fewer pulmonary complications, and shorter hospital stays while improving lymph node dissection around the recurrent laryngeal nerve without increasing paralysis rates [12,56]. Fujita et al. [81] reported a significantly lower incidence of recurrent laryngeal nerve paralysis in RAMIE than in MIE (8% vs. 34%). Seto et al. [83] demonstrated the feasibility of robot-assisted mediastinoscopic surgery in patients with poor performance status or pulmonary function.

The technical advantages of RAS, including precise forceps manipulation in confined spaces, the use of a third arm for surgical field stabilization, countertraction, and 3D magnification, have been linked to reduced blood loss and lower postoperative complication rates compared with those in conventional thoracoscopic and mediastinoscopic MIS [56,57,79,80,81,82]. Warner et al. [31] concluded that MIE remains a viable surgical option for resectable cases following neoadjuvant chemoradiotherapy, without increasing morbidity or mortality. Defize et al. [32] highlighted RAMIE’s potential as a salvage or conversion surgery for initially unresectable tumors infiltrating adjacent organs. Given the continued evolution of multidisciplinary cancer treatment, robotic MIS is expected to gain prominence for enhancing resectability and prolonging survival.

### 3.2. Evidence for MIS in GC

Table 2 presents studies evaluating the efficacy of LG compared with open gastrectomy (OG) in Asia. In Japan and Korea, randomized controlled trials (JCOG0912 and KLASS01) have demonstrated the noninferiority of laparoscopic distal gastrectomy relative to open distal gastrectomy for clinical stage I GC [37,38,40,41]. Furthermore, the feasibility of laparoscopy-assisted total or proximal gastrectomy has been validated in a single-arm, confirmatory clinical trial (JCOG1401) [39]. Although survival data from this trial have not been explicitly reported, laparoscopic total or proximal gastrectomy is weakly recommended based on extrapolated survival data from the JCOG0912 trial, which established the noninferiority of LG to OG for clinical stage I GC [37,38]. Consequently, LG for clinical stage I GC is recognized as a standard treatment option in the current Japanese guidelines [84].

For advanced GC, large-scale randomized clinical trials assessing the safety and long-term survival outcomes of laparoscopic distal gastrectomy have been conducted in Japan, Korea, and China (JLSSG0901, KLASS-02, and CLASS-01, respectively) [43,44,45,46,47]. The KLASS-02 and CLASS-01 trials confirmed the safety and noninferiority of laparoscopic versus open distal gastrectomy. However, evidence supporting the effectiveness of LG for advanced GC at clinical stage II or higher remains inconclusive [43,44,45,46,47,48].

Historically, extensive surgical approaches for advanced GC have yielded unfavorable oncological outcomes, and they are no longer recommended in the current guidelines. The JCOG9501 trial, conducted by Sasako et al. [36], demonstrated that gastrectomy with D2 lymphadenectomy and para-aortic nodal dissection did not improve survival in patients with curable GC compared with D2 lymphadenectomy alone. Another randomized controlled trial (JCOG9502) assessed the superiority of the left thoracoabdominal approach over the abdominal–transhiatal approach for GC of the cardia or subcardia; however, the findings indicated no prognostic improvement due to an increased incidence of respiratory complications [85]. Sano et al. [34] conducted the JCOG0110 trial, which revealed that D2 lymphadenectomy with splenectomy for upper GC without greater curvature invasion did not improve prognosis compared with spleen-preserving D2 dissection. Additionally, the JCOG1001 trial by Kurokawa et al. [35] found no survival benefit of gastrectomy with bursectomy over omentectomy alone in cases of resectable cT3-4 GC. Further studies are required to determine how these findings, predominantly from open surgery studies, translate to MIS and RAS.

### 3.3. RAS for GC

One of the major concerns in LG is the incidence of pancreatic fistula due to manipulation of forceps [42,49]. To address this issue and other limitations of LG, clinical trials are currently investigating the efficacy of RG compared with LG [7,53,54,55,59,63,65,66,67,68,69,70,71,72,73,74]. RAS has been demonstrated to significantly reduce the incidence of postoperative pancreatic fistula (0–3.7%). Uyama et al. [54] reported that the overall complication rate in RG was significantly lower than that in LG (2.45% vs. 6.40%, *p* = 0.0018).

A systematic review by Shibasaki et al. [86] showed that, compared with LG, RG is associated with reduced intraoperative blood loss, shorter lengths of hospital stays, and a shorter learning curve, while mortality rates remain similar. However, RG is characterized by longer operative durations and higher cost. Although morbidity and long-term outcomes appear comparable, RG exhibits potential advantages and may be applied to all patients with GC meeting LG indications. Table 3 provides an overview of randomized controlled and cohort studies assessing RG versus LG. Several studies have reported that RAS is associated with a lower conversion rate to laparotomy compared with LG. Moreover, RAS demonstrates favorable operability and safety profiles in patients with both early and advanced GC [6,53,65,72,73,74].

Recent reports have highlighted the use of RAS for advanced GC surgery in complex procedures such as para-aortic lymph nodal dissection, spleen-preserving splenic hilar nodal dissection, and esophagogastric junction cancer surgery [87,88]. It is anticipated that RAS will mitigate the negative effects associated with extensive surgery. Kinoshita et al. [88] are conducting the JCOG1809 trial, a single-arm phase II study investigating the safety and feasibility of laparoscopic or robot-assisted spleen-preserving hilar nodal dissection in advanced GC cases (cT2-4a, any N, M0) with greater curvature invasion. The study’s primary endpoint is the incidence of pancreatic fistula and/or intraperitoneal abscess, while secondary endpoints include the number of retrieved No. 10 lymph nodes, 5-year survival rate, and conversion to open surgery or splenectomy. In Japan, the JCOG1907 trial (MONALISA study), the first multicenter randomized controlled trial assessing the safety of RG across all GC stages, is currently underway [55].

### 3.4. Effect of Complications on the Prognosis of Upper Gastrointestinal Cancer

Postoperative complications in cancer surgery have been associated with prolonged hospital stays, nutritional deterioration, and significantly impaired long-term prognosis [89,90,91]. Inflammatory cytokines and chemokines, such as interleukin (IL)-6 and IL-8, have been implicated in tumor progression by promoting cancer cell proliferation and suppressing natural killer cell function. The perioperative elevation of inflammatory cytokines, which is intensified by postoperative complications, may play a crucial role in long-term prognosis [92,93]. These findings underscore the importance of preventing perioperative complications to improve oncological outcomes.

Multiple reports have described the effect of surgery and reconstruction methods for upper gastrointestinal cancer on postoperative nutritional status and quality of life (QOL). Studies, such as the Post Gastrectomy Syndrome Assessment Study (PGSAS), have assessed how the extent of gastrectomy and the reconstruction method influence postoperative QOL, nutritional status, and weight loss rates. These studies underscore the importance of minimizing surgical invasiveness and reducing complications [94,95]. Furthermore, given that malnutrition and QOL deterioration following EC surgery can significantly affect patients regardless of the presence of complications, reducing surgical invasiveness is crucial. The potential benefits of RAS in this regard may be synergistic [96].

## 4. Current Status and Future Applications of Robot-Assisted Surgery

The key advantages of RAS include enhanced microanatomy visualization through 3D magnification, precise robotic arm manipulation with a wide range of motion due to multi-joint functionality, image stabilization, and surgeon-centered control of the operative field using a third arm. The da Vinci Xi^®^ system, which has four independent and identical robotic arms, provides surgeons with the flexibility to reposition and use instruments and endoscopy as needed. Additionally, RAS has been reported to impose less physical strain on surgeons than laparoscopic surgery, which demands prolonged static muscle activity and a high physical workload [9]. These findings suggest that RAS has broad applications, including its potential use in telesurgery. As RAS becomes more widespread, it may facilitate access to cancer treatment in underserved areas.

### 4.1. Potential of Robotic Surgery for Remnant GC

In Asia, where GC is prevalent, remnant GC incidence is 0–14.3%. Iwata et al. [97] found that, during a median follow-up of 52.8 months, remnant GC incidence was 1.8% after distal gastrectomy and 6.6% after proximal gastrectomy. Multi-institutional studies by Ishida et al. and Ishizu et al. [98,99] reported incidences of 5.7% and 14.3% over 5 and 9 years, respectively. Moreover, the incidence of remnant GC increases over time after the initial gastrectomy [100]. As the aging population grows, the number of polysurgery cases is expected to rise, with increasing reports of MIS for remnant GC [15,16,17,18,19,20,21,22,23,24,25,26,27,28,29,30,101].

Compared with primary upper GC, remnant GC is associated with a lower incidence of perigastric lymph node metastasis but a higher metastatic rate to the inferior mediastinal or jejunal mesenteric lymph nodes [102]. This finding suggests a significantly altered lymphatic drainage pattern in remnant GC, necessitating thorough lymph node dissection. Nagai et al. and Tsunoda et al. reported favorable short-term outcomes of laparoscopic surgery for remnant GC, while Umeki et al. suggested that MIS may offer long-term benefits (Table 4) [17,19,29]. Our experience with RAS for remnant GC cases also supports its feasibility and safety (Figure 1A,B).

For gastric conduit cancer arising in reconstructed gastric conduits following EC surgery, early-stage cases can be managed with endoscopic submucosal dissection [103,104,105]. However, surgical intervention is necessary for advanced cases undetected by screening, and RAS is a viable option [105,106]. In one of our patients with advanced gastric conduit cancer extending through the entire reconstructed gastric conduit in the mediastinal route, curative resection via a right transthoracic approach was achieved safely. The 3D magnification and enhanced precision of forceps manipulation were particularly advantageous for dissecting dense adhesions around the lung, chest wall, tracheal membrane, and gastric conduit. We successfully performed curative total remnant gastrectomy with mediastinal lymph node dissection without severe intraoperative injuries, demonstrating the efficacy of RAS in intrathoracic reoperations (Figure 1C–F).

### 4.2. Potential of Robotic Surgery for Conversion and Salvage Surgery

Recent reports have highlighted the benefits of conversion surgery after induction chemotherapy and chemoradiotherapy for initially unresectable advanced GC and EC [107,108,109]. The REGATTA trial did not demonstrate a survival benefit for palliative gastrectomy along with chemotherapy over chemotherapy alone in patients with stage IV GC [110]. However, several studies have reported improved survival rates following palliative surgery for stage IV GC. Min et al. compared short- and long-term outcomes between laparoscopic and open radical gastrectomy with maximal metastasectomy in patients with stage IV GC. They reported a postoperative complication rate of 5.6% for radical LG with maximal metastatic resection, which was significantly lower than the 23.4% observed in the open surgery group. Furthermore, the 2-year overall survival (OS) rate and median survival time (MST) in Min et al.’s study—55.6% and 26.8 months, respectively—were markedly higher than the 2-year OS rate of 25.1% and MST of 14.3 months reported in the REGATTA trial [108]. The favorable long-term outcomes observed up to 2 years after surgery can be attributed to the minimal invasiveness of MIS and maximal radical resection of lesions.

Hisamori et al. [109] also reported outcomes of LG in highly advanced GC, with 5-year progression-free survival (PFS) and 5-year OS rates of 9.1% and 27.3%, respectively, in patients with cStage IV GC. These findings closely align with the 5-year OS rate of 23.4% reported by Min et al., further supporting the efficacy of MIS in palliative gastrectomy. Ojima et al. [36] explored the potential of reducing the invasiveness of metastasectomy through RAS in para-aortic lymphadenectomy, an approach not previously shown to be beneficial in JCOG9501. This method could further improve prognosis in conversion surgery cases following first-line chemotherapy for patients with stage IV GC, as reported by Yoshida et al. [87,107].

For resectable advanced GC, several neoadjuvant chemotherapy (NAC) regimens have demonstrated efficacy in extending OS and PFS. These include the S-1 plus oxaliplatin (SOX) regimen under evaluation in JCOG1509 [111], docetaxel-enhanced SOX (DOS) regimen reported by Kang et al. [112], and 5-fluorouracil-leucovorin-oxaliplatin-docetaxel (FLOT4) regimen [113]. Additionally, studies such as ATTRACTION-4 [114], KEYNOTE-859 [115], and CheckMate-649 [116] have provided compelling evidence for combining chemotherapy with immune checkpoint inhibitors in unresectable GC and EC. These advancements have significantly improved response rates, extended OS and PFS, and increased the likelihood of transition to conversion surgery. The integration of multidisciplinary treatments, including MIS and RAS in conjunction with drug therapy, may mitigate the historically negative effect of extensive surgery for advanced GC [107].

### 4.3. RAS in Polysurgery Cases

The use of RAS in polysurgery cases offers several advantages, particularly in securing port insertion sites by detaching adhesions from the abdominal wall and between organs. To minimize invasiveness, preoperative computed tomography (CT) imaging can help assess the alignment of the abdominal wall and intestines across different phases, enabling surgeons to determine adhesion-free areas for port placement. Additionally, intra-abdominal CT evaluation can help identify the safest initial port insertion site. Following the first port placement, laparoscopic and robotic visualization allows for optimal positioning of the secondary ports, facilitating adhesion dissection and further port insertions in space-restricted environments (Figure 2A,B).

After confirming the altered microanatomy caused by fibrous scars between the target organ and surrounding structures, it is crucial to restore the normal anatomy before proceeding with cancer surgery. This procedure requires careful visual inspection, precise sharp dissection, and controlled coagulation to ensure the safe separation of adhesions. Given the increasing number of cancer survivors and the aging population, the incidence of polysurgery cases is expected to increase [14]. Even in postoperative patients with an ileal conduit stoma who have undergone bladder cancer surgery, intraperitoneal observation following the insertion of the initial port allows for the strategic placement of subsequent ports. RAS facilitates procedural flexibility, as it enables surgical maneuvers independent of port placement (Figure 2C,D). With innovative techniques, the application of RAS can be expanded, contributing to advancements in cancer treatment.

## 5. Challenges and Developments in RAS Systems

One of the primary disadvantages of RAS is the risk of crush injury at the port insertion site. However, newer systems, such as the Senhance^®^ surgical system (TransEnterix Surgical Inc., Morrisville, USA), have been developed to mitigate these risks. Additionally, to minimize interference and collisions between robotic arms and surgical instruments, manufacturers are focusing on improvements such as high-performance articulated forceps, miniaturization of robotic components, and integration of independent forceps arm carts. Systems like the Hugo^™^ RAS (Medtronic, Minneapolis, USA) and Senhance^®^ have already adopted independent forceps arm cart models.

Further advancements include the development of surgical tools in high demand, such as articulated ultrasonic coagulation and incision devices, high-precision automatic suturing systems, and sealing devices. These innovations are expected to be implemented in clinical practice. Although RAS offers several advantages over conventional surgical techniques, the absence of intraoperative force feedback—combined with the potential for excessive grasping forces—can lead to unintended tissue damage. This limitation makes it difficult for the surgeon to accurately gauge the appropriate amount of force required during manipulation. Additionally, the integration of tactile feedback into robotic forceps is anticipated to enhance precision and reduce the risk of inadvertent tissue damage. Notably, the force feedback function of the Saroa^®^ system (RIVERFIELD Inc., Tokyo, Japan) and the expected tactile sensation reproduction capabilities of the next-generation da Vinci 5^®^ (Intuitive Surgical, Sunnyvale, CA, USA) are being closely monitored for their potential contributions to surgical safety [117].

Technological advancements are also being made in imaging quality, artificial intelligence (AI)-driven applications, measurement capabilities, 3D imaging, anatomical reconstructions, and virtual reality integration. The integration of AI and machine learning (ML) into RAS is expected to enhance intraoperative decision-making by improving the identification of fine and complex anatomical structures. These technological advancements may improve short-term surgical outcomes. However, several challenges remain, including the high cost of RAS, complexity of system maintenance, and large physical footprint of the equipment. Future innovations—such as AI-driven automation, nanorobots, microdissection techniques, semi-automated remote robotic systems, and the application of 5G connectivity for remote surgery—have the potential to further expand the clinical utility of RAS. The development of telesurgery capabilities may allow the introduction of RAS to medically underserved regions, thereby helping bridge the evidence–practice gap in cancer treatment, reduce delays in surgical skill acquisition, and mitigate the global shortage of trained surgeons [118,119,120]. Although a few companies have attempted to introduce systems capable of competing with the dominance of the DVSS, none have yet achieved comparable performance [8]. However, by leveraging the strengths of each available system, RAS can significantly improve the outcomes of MIS for upper gastrointestinal cancers. Although current applications are limited, continued technological refinement is expected to accelerate the expansion of RAS capabilities.

From a training perspective, the learning curve of RAS is shorter than that of laparoscopic surgery [10,11,12]. Sun et al. [121] reported that the learning curve for LG was overcome after approximately 20 cases, whereas only 10 cases were required for RG. In contrast, Kim et al. [122] found that 25 cases were required to surpass the learning curve and achieve sufficient proficiency in RG. Both studies indicated that prior LG experience and the mode of training influence the progression through the learning curve [121,122]. Additionally, Zheng et al. [123] reported that 22 cases were required to overcome the learning curve for robotic distal gastrectomy and 20 cases for robotic total gastrectomy, suggesting that proficiency levels may vary slightly between different robotic procedures. In Japan, the Proctor system and certification requirements for robotic surgeons contribute to improved surgical precision. Studies have shown that robotic GC surgeries performed by young specialists achieve satisfactory outcomes after approximately 20 procedures [86,124]. To further enhance surgical safety during the initial learning phase, emphasis has been placed on annotation tools, dual-console systems, and “role-sharing surgery,” in which assistants contribute not only to surgical field development but also to dissection tasks.

The DVSS includes a proficiency tracking feature that records surgeon performance. By using real-time data, surgical videos, and educational content, surgeons can refine their techniques more effectively. Additionally, the SimNow^™^ simulator, developed by Intuitive Surgical, enables robotic surgery training without direct patient interaction. With further advancements in simulator systems, RAS training is expected to progress rapidly. Moreover, cadaver training programs complement traditional animal laboratory training by providing conditions closely replicating real surgical scenarios. These improvements are facilitating skill acquisition at an accelerated pace.

### Limitations and Barriers to RAS Implementation

Several challenges must be addressed to broaden RAS adoption. First, in cases of polysurgery, open surgery may still be required for patients with a history of refractory bowel obstruction or those in whom preoperative CT imaging suggests that initial port placement is not feasible. For patients with severe adhesions, RAS remains a less commonly viable minimally invasive option.

Second, the cost and facility requirements for RAS implementation pose significant barriers. Gao et al. [72] and Sakai et al. [125] reported that the increased use of RAS does not necessarily result in higher hospital revenue than conventional laparoscopic surgery. Sakai et al. conducted a comparative cost analysis of robot-assisted versus laparoscopic surgery to evaluate the financial impact of each procedure. They found that RAS generally yielded lower profits than laparoscopic procedures, with robot-assisted proximal gastrectomy even resulting in a negative gross surgical profit. The high costs of robotic instruments and ongoing maintenance were not adequately offset by government-determined reimbursement rates, leading to financial losses in these cases. Additionally, they noted that more complex procedures, such as esophageal, rectal, bladder, and prostate surgeries, generated higher gross profits. The financial feasibility of RAS may vary depending on the healthcare system and the specific surgical procedure, with potential economic benefits emerging in the future. Currently, robotic systems are primarily introduced in high-volume centers and university hospitals, raising concerns about accessibility to small- to medium-sized hospitals. However, the Hinotori^®^ system (Medicaroid Corporation, Tokyo, Japan), a compact and cost-effective alternative, is expected to facilitate wider adoption. Priced at approximately half to two-thirds the cost of the da Vinci Xi, Hinotori offers an interface similar to that of DVSS, making it an attractive option for institutions already familiar with the da Vinci systems.

Third, the centralization of RAS facilities and patient selection is another critical consideration. However, RAS can be effectively implemented using appropriate training programs. A nationwide propensity score-matching analysis using the National Clinical Database indicated that RAS, after insurance approval, demonstrated safety comparable with LG across various institutions [6]. This suggests that the expansion of robotic applications in surgical oncology is both feasible and beneficial.

Finally, financial constraints related to insurance reimbursement remain a major challenge. The high cost of RAS currently limits their widespread adoption; however, the emergence of cost-effective robotic systems may help address this issue. If hospitals recognize the financial benefits of RAS, broader implementation is likely to follow.

## 6. Conclusions and Future Directions

Recent advancements in MIS and RAS can improve the outcomes of upper gastrointestinal cancer surgery by reducing invasiveness and improving oncological outcomes. Previous studies have shown that MIS is a viable option for polysurgery cases, metachronous multiple cancers, and salvage or conversion surgeries. Among these, RAS is particularly advantageous because of its potential to improve short-term postoperative outcomes and overall prognosis.

Further research and data collection are necessary to establish robust evidence supporting the expanded application of RAS in gastrointestinal cancer surgery. With continued technological and procedural developments, RAS is poised to play an increasingly significant role in the future of oncologic surgery.

## Figures and Tables

**Figure 1 cancers-17-01933-f001:**
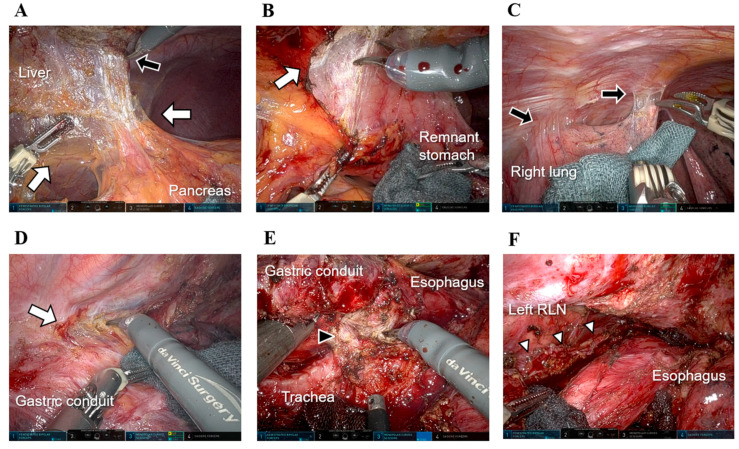
Intraoperative views of robot-assisted remnant gastrectomy. (**A**) A patient with remnant gastric cancer following distal gastrectomy presented with dense fibrous adhesions in the abdominal wall, liver, pancreas, and remnant stomach. Robotic manipulation was used to carefully dissect the extensive adhesions between the liver surface (black arrow) and pancreas (white arrows). (**B**) Using a 3D magnified view, robotic sharp and blunt dissections were performed to separate the dense adhesions between the liver surface and remnant stomach (white arrows). (**C**) In a patient with gastric conduit cancer and a history of transthoracic esophagectomy with posterior mediastinal reconstruction, robot-assisted resection of extensive adhesions between the thoracic wall and right lung provided an adequate view of the surgical field (black arrows). (**D**) Robotic dissection was used to safely separate firm adhesions involving the thoracic wall, descending aorta, and reconstructed gastric conduit, thereby minimizing the risk of traumatic injury or hemorrhage (white arrow). (**E**) Dense scarring is noted around the esophagogastric anastomosis. Robotic manipulation enabled precise dissection of adhesions between the membranous portion of the trachea and the reconstructed gastric conduit (black arrowhead). (**F**) The left recurrent laryngeal nerve was meticulously preserved along its entire course, and dissection of the surrounding structures was completed (white arrowheads). RLN, recurrent laryngeal nerve.

**Figure 2 cancers-17-01933-f002:**
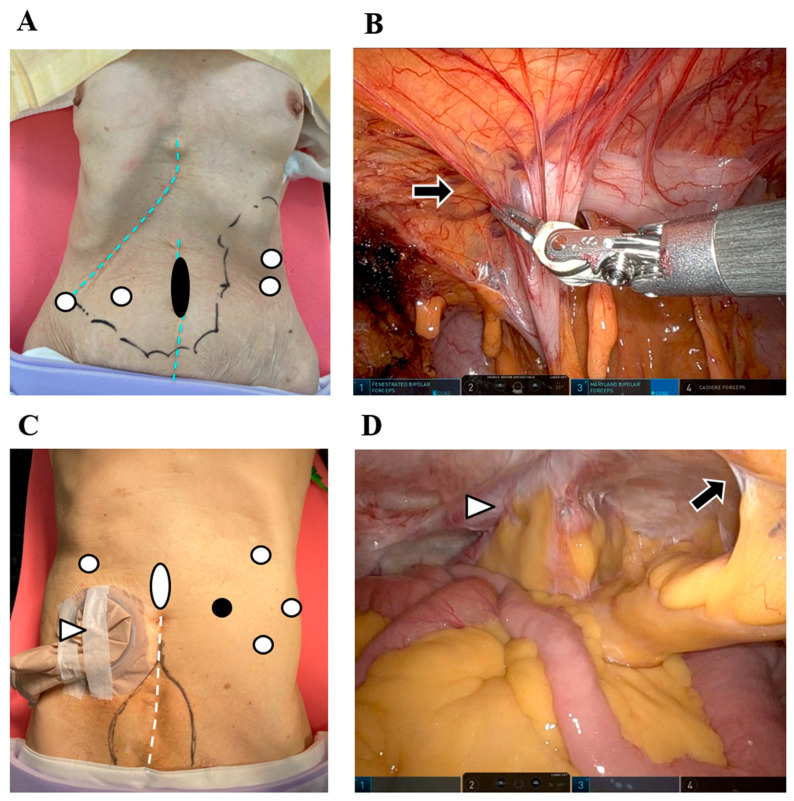
Port placement and intraoperative views in patients with prior surgeries for gastric cancer. (**A**) In a postoperative patient with a history of liver resection and right hemicolectomy for metachronous liver and colon cancer, port placement was planned based on previous surgical scars and preoperative computed tomography (CT) images (sky blue dotted lines: prior surgical scars; black spot: first port site; white spots: second, third, fourth, and access port sites). (**B**) During the insertion of the fourth robotic port, robot-assisted dissection was performed to carefully separate the extensive adhesions between the abdominal wall and the digestive tract (black arrow). (**C**) In a postoperative patient with an ileal conduit stoma following bladder cancer surgery, port placement was planned to avoid both the stoma and previous incision site (white arrowhead: stoma; white dotted line: previous surgical scar; black spot: first port site; white spots: second, third, fourth, and access port sites; black line: abdominal wall incisional hernia). (**D**) Intra-abdominal observation during robotic port insertion helped prevent injury to the stoma and incisional hernia (black arrow: adhesion between the abdominal wall and omentum; white arrowhead: stoma).

**Table 1 cancers-17-01933-t001:** Summary of studies evaluating thoracoscopic and robot-assisted minimally invasive esophagectomy in patients with esophageal cancer.

Author	Year	Design	Procedure	No. of Cases	Conversion to Open Surgery	Thoracic Operative Time (min)	Blood Loss (mL)	No. of Retrieved LNs	Complications
Osugi. [50]	2003	Retrospective	MIE	77	NR	227	284	33.9	Pneumonia 15.6% RLNP 14.3%
			OE	72	NA	186	310	32.8	Pneumonia 19.4% RLNP 12.5%
Biere [51]	2012	RCT	MIE	59	14%	329	200	20	Pneumonia 9% RLNP 2%
			OE	56	NA	299	475	21	Pneumonia 29% RLNP 14%
Weksler [75]	2012	Retrospective	RAMIE	569	6.7%	NR	NR	16.0	NR
		PSM	MIE	569	12.0%	NR	NR	16.0	NR
		NDB	OE	569	NA	NR	NR	13.0	NR
Van der Sluis [56]	2017	RCT	RAMIE	54	2%	170	120	27	Pneumonia 28% RLNP 9%
De Groot [57]	2020		OE	55	NA	134	200	25	Pneumonia 55% RLNP 11%
Sarkaria [76]	2019	Prospective	RAMIE	64	NR	384	250	25	Pneumonia 14.1% RLNP 3.1%
			OE	106	NA	326	350	22	Pneumonia 34% RLNP 0%
Zhang [77]	2019	Retrospective	RAMIE	76	2.6%	303.5	200	19.7	Pneumonia 6.6% RLNP 6.6%
			MIE	108	0	277.2	200	20.3	Pneumonia 9.3% RLNP 6.5%
Yun [78]	2020	Retrospective	RAMIE	130	2.3%	275.6	110.8	39.1	Pneumonia 3.8% RLNP 25.4%
		PSM	OE	241	NA	240.0	93.8	38.3	Pneumonia 10.8% RLNP 19.9%
Tagkalos [3]	2020	Prospective	RAMIE	50	NA	223	331	27	Pneumonia 18%
			MIE	50	NA	202	350	23	Pneumonia 12%
Gong [80]	2020	Retrospective	RAMIE	91	NR	318.0	215	22.8 (Upper mediastinal LN 6.2)	Pneumonia 9.9% RLNP 22.0%
			MIE	144	NR	321.1	200	23.1 (Upper mediastinal LN 5.6)	Pneumonia 10.4% RLNP 23.6%
			OE	74	NA	299.4	290	24.1 (Upper mediastinal LN 4.3)	Pneumonia 13.0% RLNP 15.6%
Yang [79]	2022	RCT	RAMIE	181	7	203.8	200	15	Pneumonia 9.9% RLNP 32.6%
			MIE	177	6	244.9	200	14	Pneumonia 11.9% RLNP 27.1%
Fujita [81]	2022	Retrospective	RAMIE	50	0	Total 448.1	Total 111.6	NR	Pneumonia 8.0% RLNP 8.0%
		PSM	MIE	50	0	Total 383.6	Total 153.5	NR	Pneumonia 12.0% RLNP 34.0%

Abbreviations: LN, lymph node; MIE, minimally invasive esophagectomy; NA, not available; NDB, national database; NR, not reported; OE, open esophagectomy; PSM, propensity score matching; RAMIE, robot-assisted minimally invasive esophagectomy; RCT, randomized controlled trial; RLNP, recurrent laryngeal nerve paralysis.

**Table 2 cancers-17-01933-t002:** Summary of studies comparing open gastrectomy and laparoscopic gastrectomy for gastric cancer.

Author	Year	Stage Design	Procedure	No. of Cases	Conversion to Open Surgery (%)	Mean Operative Time (min)	Mean Blood Loss (mL)	Retrieved Lymph Node	Postoperative Hospital Stay (Days)	Overall Morbidity (%)	5 yr RFS	5 yr OS
Katai [37] Katai [38] JCOG0912	2017 2020	I RCT	LDG	457	3.5	278	38	39	11.3	3.3 (C–D III≦)	95.1%	NR
			ODG	455	NA	194	115	39	24.9	3.7 (C–D III≦)	94.0%	NR
Kim [40] Kim [41] KLASS-01	2016 2019	I RCT	LDG	644	0.9	184.1	190.6	40.5	7.1	13.0	97.1%	94.2%
			ODG	612	NA	139.4	110.8	43.7	7.9	19.9	97.2%	93.3%
Yoshida [42]	2017	I NDB PSM	LDG	14,386	47.1	287	50	NR	12	SSSI 1.0% PF 1.0%	NR	NR
			ODG	14,386	NA	209	185	NR	15	SSSI 1.9% PF 0.8%	NR	NR
Hu [43] Huang [44] CLASS-01	2016 2022	I-IV RCT	LDG	519	6.4	217.3	105.5	36.1	10.8	15.2	3 yr 76.6%	72.6%
			ODG	520	NA	186.0	117.3	36.9	11.3	12.9	3 yr 77.8%	76.3%
Lee [45] Son [46] KLASS-02	2019 2022	IB-III RCT	LDG	460	3.7	225.7	138.3	46.6	8.1	22.0	79.5%	88.9%
			ODG	458	NA	162.3	222.0	46.9	9.1	24.5	81.1%	88.7%
Etoh [47] JLSSG0901	2023	IB-III RCT	LDG	227	1.2	205	30	43	NR	11.5	75.7%	81.7%
			ODG	233	NA	291	141	43	NR	10.7	73.9%	79.8%
Kinoshita [48] LOC-A Study	2019	II-III PSM	LG	305	1.3	365	140	43	12	20.1	Recurrence rate 29.8%	54.2%
			OG	305	NA	228	396	34	12	18.7	Recurrence rate 30.8%	53.0%
Yoshida [42]	2017	IIA-IV NDB PSM	LDG	3738	47.1	296	50	NR	13	NS	NR	NR
			ODG	3738	NA	222	240	NR	15		NR	NR

Abbreviations: C–D classification, Clavien–Dindo classification; LDG, laparoscopic distal gastrectomy; LG, laparoscopic gastrectomy; NA, not available; NDB, national database; NR, not reported; NS, not significant; ODG, open distal gastrectomy; OG, open gastrectomy; OS, overall survival; PF, pancreatic fistula; PSM, propensity score matching; RCT, randomized controlled trial; RFS, relapse-free survival; SSSI, superficial surgical site infection.

**Table 3 cancers-17-01933-t003:** Summary of studies comparing laparoscopic gastrectomy and robotic gastrectomy for gastric cancer.

Author	Year	Design	Procedure	No. of Cases	Stage ≥ II	TG or PG (%)	Conversion to Open Surgery	Operative Time (min)	Blood Loss (mL)	Retrieved Lymph Node	Postoperative Hospital Stay (Days)	Overall Morbidity (%)	RFS (%)	OS (%)
Wang [59], China	2016	RCT	RG	151	76	37	1.9%	243	94	30.1	5.6	10.3	NR	NR
			OG	145	79	31	NA	192	153	29.1	6.7	9.3	NR	NR
Pan [60], China	2017	RCT	RG	102	78.0	64.7	0	153	41	36.1	3.8	5.0	NR	NR
			LG	61	89.0	73.8	0	152	84	30.0	5.4	19.7	NR	NR
Lu J [61], China	2021	RCT	RG	141	NR	0	NR	201	41	17.6	7.9	9.2	NR	NR
			LG	142	NR	0	NR	182	56	15.8	8.2	17.6	NR	NR
Ojima [7], Japan	2021	RCT	RG	117	41.9	40.7	3.4	297	25	35	12	8.8	NR	NR
			LG	119	40.3	31.6	1.7	245	25	30	13	19.7	NR	NR
Kim [62], South Korea	2016	Prospective	RG	185	18.9	16.2	1.1	221	50	34	6	11.9	NR	NR
			LG	185	10.2	16.2	0.5	178	55	32	6	10.3	NR	NR
Uyama [54], Japan	2019	Prospective,	RG	326	12	22	0.3	313	20	38.5	9	41.1	NR	NR
Okabe [63], Japan	2019	Prospective,	RG	115	40.9	37.4	1.7	372	15	46	12	9.6	NR	NR
Tokunaga [64], Japan	2016	Prospective	RG	120	1	12	2.5	349	19	NR	9	14.2	NR	NR
Parisi [65], Italy	2017	Retrospective, PSM	RG	151	44	26	4.6%	365	118	27.8	8.9	17.9	NR	NR
			LG	151	44	32	5.3%	220	96	24.6	9.1	11.9	NR	NR
			OG	302	53	32	NA	199	127	25.8	12.7	19.5	NR	NR
Wang [66], China	2019	Retrospective, PSM	RG	253	76	43	NR	242	149	NR	10.2	18.8	NR	NR
			LG	253	76	44	NR	238	144	NR	11.6	24.5	NR	NR
Ryan [67], USA	2020	Retrospective	RG	631	66	28	NR	NR	NR	19.6	10.2	NR	NR	MST 56.2 mo.
			LG	1262	66	28	NR	NR	NR	17.4	11.6	NR	NR	MST 49.2 mo.
Shibasaki [68], Japan	2020	Retrospective, PSM	RG	354	38	30	0	360	37	37	12	3.7 (C–D III ≦)	NR	NR
			LG	354	37	29	0.1	347	28	36	13	7.6 (C–D III ≦)	NR	NR
Hikage [53], Japan	2021	Retrospective, PSM	RG	342	4.7	16	2.0	321	15	42.0	8	13.2	5 yr 95.2	5 yr 96.4
			LG	342	7.0	15	2.5	282	14	40.5	9	18.4	5 yr 93.4	5 yr 94.8
Suda [6], Japan	2022	Retrospective, NCD, PSM	RG	2671	NA	14.5	0.3	354	20	NR	10	4.9(C–D III ≦)	NR	NR
			LG	2671	NA	14.5	0.5	268	15	NR	11	3.9 (C–D III ≦)	NR	NR
Shimoike [69], Japan	2022	Retrospective	RG	336	33	24	0	370	0	NR	10	14.9 (C–D II ≦)	NR	NR
Omori [70], Japan	2022	Retrospective, PSM	RG	210	48	32	NR	208	13	NR	7	1.0	NR	NR
			LG	210	48	35	NR	231	42	NR	8	4.8	NR	NR
Tian [71], China	2022	Retrospective, PSM	RG	463	65	20	NR	205	74	32.2	7.3	2.7	3 yr 77.0	3 yr 81.2
			LG	877	68	21	NR	185	78	30.8	7.6	3.2	3 yr 77.0	3 yr 80.3
Gao [72], China	2022	Retrospective, PSM	RG	410	88	0	0.6	205	139	31.4	9.0	13.7	3 yr 72.9	3 yr 75.5
			LG	410	87	0	1.4	185	167	29.4	9.1	16.6	3 yr 71.4	3 yr 73.1
Li [73], China	2023	Retrospective, PSM	RG	1776	64.6	30.7	1.2	249	127	32.5	9.2	12.6	5 yr 79.8	5 yr 80.8
			LG	1776	65.0	30.7	1.6	220	143	30.7	9.3	15.2	5 yr 78.5	5 yr 79.5
Lu [74], China	2024	Retrospective, PSM	RG	1034	61.9	34.3	0.4	223	98	30.8	9.4	12.2	3 yr 77.4	3 yr 79.7
			LG	1034	61.6	33.1	1.5	210	118	30.8	9.9	12.1	3 yr 76.7	3 yr 78.4

Abbreviations: C–D classification, Clavien–Dindo classification; LG, laparoscopic gastrectomy; MST, median survival time; NA, not available; NR, not reported; OG, open gastrectomy; OS, overall survival; PG, proximal gastrectomy; PSM, propensity score matching; RCT, randomized controlled trial; RFS, relapse-free survival; RG, robotic gastrectomy; TG, total gastrectomy.

**Table 4 cancers-17-01933-t004:** Summary of studies evaluating laparoscopic completion gastrectomy for remnant gastric cancer.

Author	Year	Procedure	No. of Cases	Stage (Early/Advanced)	Conversion to Open Surgery	Mean Operative Time (min)	Mean Blood Loss (mL)	Retrieved Lymph Node	Postoperative Hospital Stay (Days)	Complications	RFS (%)	OS (%)
Qian F [15]	2010	Lap	15	0/15	6.7	205	110	18	13	6.7	NR	NR
Kim [16]	2014	Lap	17	13/4	0	197.2	NR	12.9	11.1	23.5	NR	NR
		Open	50	NR	NA	149.3	NR	NR	13.8	30.0	NR	NR
Nagai [17]	2014	Lap	12	10/2	0	362.3	68.5	23.7	11.3	0	NR	3 yr 77.8
		Open		6/4	NA	270.5	746.3	15.7	24.9	20	NR	3 yr 100 5 yr 72.9
Son [18]	2015	Lap		11/6	47.1	234.4	227.6	18.8	9.3	35.2	NR	5 yr 66.7
		Open		4/13	NA	170.0	184.1	22.3	9.3	29.4	NR	5 yr 60.3
Tsunoda [19]	2016	Lap	10	9/1	0	325	55	22	13	10	NR	NR
Otsuka [20]	2019	Lap	7	6/1	0	364	70	22	13.4	28.6	NR	NR
		Open	20	12/8	NA	309	1066	12	16	50	NR	NR
Booka [21]	2019	Lap	8	2/6	25.0	307.5	135.5	8.8	10.6	37.5	NR	NR
		Open	23	8/15	NA	295.8	568.3	6	21.3	26.1	NR	NR
Kaihara [22]	2019	Lap	6	2/4	16.7	310.5	50	7	9	50.0 (C–D II ≦)	NR	5 yr 80.0
		Open		5/10	NA	263	465	3	9	33.3 (C–D II ≦)	NR	5 yr 60.6
Ota [23]	2020	LG		2/5	0	397	70	21	30	13.3	NR	3 yr 100
		OG	15	11/4	NA	271	245	8	23	20.0	NR	3 yr 86.7
Kitadani [24]	2020	LG	23	18/5	13.0	302	115	8	11	21.7	5 yr 87	5 yr 62
		OG	15	5/10	NA	281	290	12	14	40.0	5 yr 78	5 yr 77
Albossani [25]	2020	LG	30	24/6	13.3	225	166	16	9.5	37.0	NR	NR
		RG	25	18/7	0	292	202	18	8.9	40.0	NR	NR
Li [26]	2021	LG	41	14/27	19.5	297.9	288.8	13.6	9	22.0	3 yr 57.5	3 yr 60.0
		RG	29	10/19	17.2	272	229.2	13.6	9	27.6	3 yr 65.5	3 yr 69.0
Wu [27]	2022	LG		16/20	NR	243.1	188.3	14	7.6	8.3	NR	3 yr 75.6
		OG		17/31	NA	215.7	305.8	10.7	11.2	20.8	NR	3 yr 73.3
Aoyama [28]	2023	LG	327	52/79	NR	344	Less	14	13	28.4	3 yr 71.9	3 yr 77.9
		OG	195	75/114	NA	273		10	16	47.7	3 yr 62.2	3 yr 76.2
Umeki [29]	2023	LG	46	34/12	0	311.5	35.5	13.5	16.5	8.7	3 yr 72.3	3 yr 80.2
Zhong [30]	2024	LG	46	12/34	4.3	163.9	59.7	19.2	11.9	28.0	3 yr 61.6	3 yr 56.3
		OG	160	20/140	NA	225.7	220.4	15.6	18.7	35.0	3 yr 60.8	3 yr 50.0

Abbreviations: C–D, Clavien–Dindo classification; LG, laparoscopic gastrectomy; NA, not available; NR, not reported; OG, open gastrectomy; OS, overall survival; RFS, relapse-free survival; RG, robotic gastrectomy.

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
