# Peer review of "Current Status and Future Applications of Robotic Surgery in Upper Gastrointestinal Surgery: A Narrative Review"

_cancers, 2025, doi:10.3390/cancers17121933_

Round 1

Reviewer 1 Report

Comments and Suggestions for Authors

This study explores the evolution and current role of robotic-assisted surgery (RAS) in upper gastrointestinal (GI) cancer treatment. It highlights how robotic systems, particularly the Da Vinci Surgical System (DVSS), have improved surgical precision, visualization, and minimally invasive techniques. The authors emphasize that robotic surgery is increasingly being used in gastric and esophageal cancer surgery, particularly in complex and high-risk cases. Concerning the text of the manuscript, the following issues are essential to the study:

The manuscript extensively discusses the advantages of robotic-assisted surgery (RAS) but lacks a critical assessment of its limitations in real-world clinical settings. While some drawbacks such as cost and learning curve are mentioned, a more balanced discussion is needed to avoid bias:

  1. Include comparative cost analyses between robotic, laparoscopic, and open surgery, emphasizing financial feasibility in different healthcare systems.
  2. Discuss technical challenges such as the lack of haptic feedback in robotic surgery and its impact on surgical precision.
  3. Address accessibility issues, particularly in low-resource settings, and suggest potential solutions for wider implementation
  4. To highlight and discuss how the learning curve completion is different between Robotic surgery, laparoscopic and open surgery and how it can affect the results of the robotic surgery in upper GI – operations.
Comments on the Quality of English Language

In terms of english language the manuscript has been well written. 

Author Response

We thank the reviewer for their thoughtful and constructive feedback, which has helped us improved our manuscript. Please see the detailed responses to each of the comments and the corresponding revisions made to the manuscript.

COMMENT 1.

Include comparative cost analyses between robotic, laparoscopic, and open surgery, emphasizing financial feasibility in different healthcare systems.

Reply: In the section “5.1. Limitations and barriers to RAS implementation”, we briefly mentioned the medical economic aspects of RAS by citing the paper by Sakai et al. We have added a detailed comparison of the cost-effectiveness of laparoscopic surgery and RAS as follows:

“Sakai et al. conducted a comparative cost analysis of robot-assisted versus laparoscopic surgery to evaluate the financial impact of each procedure. They found that RAS generally yielded lower profits than laparoscopic procedures, with robot-assisted proximal gastrectomy even resulting in a negative gross surgical profit. The high costs of robotic instruments and ongoing maintenance were not adequately offset by government-determined reimbursement rates, leading to financial losses in these cases. Additionally, they noted that more complex procedures, such as esophageal, rectal, bladder, and prostate surgeries, generated higher gross profits. The financial feasibility of RAS may vary depending on the healthcare system and the specific surgical procedure, with potential economic benefits emerging in the future.” (Page 14, Line 443-452)

COMMENT 2.

Discuss technical challenges such as the lack of haptic feedback in robotic surgery and its impact on surgical precision.

Reply: The following texts and reference have been added to provide additional discussion regarding the technical challenges of robotic surgery and its impact on surgical systems:
“Although RAS offers several advantages over conventional surgical techniques, the absence of intraoperative force feedback—combined with the potential for excessive grasping forces—can lead to unintended tissue damage. This limitation makes it difficult for the surgeon to accurately gauge the appropriate amount of force required during manipulation. Additionally, the integration of tactile feedback into robotic forceps is anticipated to enhance precision and reduce the risk of inadvertent tissue damage. Notably, the force feedback function of the Saroa® system (RIVERFIELD Inc., Tokyo, Japan) and the expected tactile sensation reproduction capabilities of the next-generation da Vinci 5® (Intuitive Surgical) are being closely monitored for their potential contributions to surgical safety [117].” (Page 13, Line 381-390)

Reference
117.   Y. Sakai, M. Tokunaga, Y. Yamasaki, H. Kayasuga, T. Nishihara, K. Tadano, et al. Evaluating the benefit of contact-force feedback in robotic surgery using the Saroa surgical system: A preclinical study. Asian J Endosc Surg. 2024 Oct;17(4):e13395.

COMMENT 3.

Address accessibility issues, particularly in low-resource settings, and suggest potential solutions for wider implementation

Reply: In the "5. Challenges and developments in RAS systems" section, the following citation and references have been added:
 " The integration of AI and machine learning (ML) into RAS is expected to enhance intraoperative decision-making by improving the identification of fine and complex anatomical structures. These technological advancements may improve short-term surgical outcomes. However, several challenges remain, including the high cost of RAS, complexity of system maintenance, and large physical footprint of the equipment. Future innovations—such as AI-driven automation, nanorobots, microdissection techniques, semi-automated remote robotic systems, and the application of 5G connectivity for remote surgery—have the potential to further expand the clinical utility of RAS. The development of telesurgery capabilities may allow the introduction of RAS to medically underserved regions, thereby helping bridge the evidence–practice gap in cancer treatment, reduce delays in surgical skill acquisition, and mitigate the global shortage of trained surgeons [118–120].”  (Page 13, Line 393-404)

References

  1. Chatterjee, S.; Das, S.; Ganguly, K.; Mandal, D. Advancements in robotic surgery: innovations, challenges and future prospects. J Robot Surg. 2024, 18, 28.
  2. Ebihara, Y.; Hirano, S.; Kurashima, Y.; Takano, H.; Okamura, K.; Murakami, S.; Shichinohe, T.; Morohashi, H.; Oki, E.; Hakamada, K.; et al. Tele-robotic distal gastrectomy with lymph node dissection on a cadaver. Asian J Endosc Surg. 2024, 17, e13246.
  3. Takahashi, Y.; Hakamada, K.; Morohashi, H.; Akasaka, H.; Ebihara, Y.; Oki, E.; Hirano, S.; Mori, M. Reappraisal of telesurgery in the era of high-speed, high-bandwidth, secure communications: Evaluation of surgical performance in local and remote environments. Ann Gastroenterol Surg. 2023, 7, 167-174.

COMMENT 4.

To highlight and discuss how the learning curve completion is different between Robotic surgery, laparoscopic and open surgery and how it can affect the results of the robotic surgery in upper GI – operations.

Reply: We have additionally cited the following references, which shows data from a CUSUM analysis study on the learning curve of minimally invasive gastrectomy, and have made modifications to help readers understand by describing the detailed analysis results of laparoscopic and robotic-assisted gastrectomy.

“Sun et al. [121] reported that the learning curve for LG was overcome after approximately 20 cases, whereas only 10 cases were required for RG. In contrast, Kim et al. [122] found that 25 cases were required to surpass the learning curve and achieve sufficient proficiency in RG. Both studies indicated that prior LG experience and the mode of training influence the progression through the learning curve [121, 122]. Additionally, Zheng et al. [123] reported that 22 cases were required to overcome the learning curve for robotic distal gastrectomy and 20 cases for robotic total gastrectomy, suggesting that proficiency levels may vary slightly between different robotic procedures.”  (Page 13, Line 411-418)

References

  1. Sun, L, F.; Liu, K.; Su, X, S.; Wei, X.; Chen, X, L.; Zhang, W, H.; Chen, X, Z.; Yang, K.; Zhou, Z, G.; Hu, J, K. Robot-Assisted versus Laparoscopic-Assisted Gastrectomy among Gastric Cancer Patients: A Retrospective Short-Term Analysis from a Single Institution in China. Gastroenterol Res Pract. 2019, 9059176.
  2. Kim, M,S.; Kim, W, J.; Hyung, W, J.; Kim, H, I.; Han, S, U.; Kim, Y, W.; Ryu, K, W.; Park, S. Comprehensive Learning Curve of Robotic Surgery: Discovery From a Multicenter Prospective Trial of Robotic Gastrectomy. Ann Surg. 2021, 273, 949-956.
  3. Zheng-Yan, L.; Feng, Q.; Yan, S.; Ji-Peng, L.; Qing-Chuan, Z.; Bo, T.; Rui-Zi, G.; Zhi-Guo, S.; Xia, L.; Qing, F.; et al. Learning curve of robotic distal and total gastrectomy. Br J Surg. 2021, 108, 1126-1132.

Reviewer 2 Report

Comments and Suggestions for Authors

Dear colleagues, 

congratulations for your work. It was a hard compilation of papers and their results. 
maybe i would like to find your results into the paper. 
The tables ara a l’argentí ones - it could be possible to resume .? Is difficult understsndin it

Author Response

Reply: We appreciate the reviewer for their thoughtful and constructive feedback, which has helped us improved our manuscript.

In our experience, there were some polysurgery cases, but the robotic surgical procedures for remnant gastric cancer were so diverse and we thought it would not be useful as a reference. Please forgive us for not including our experiences of robotic remnant gastric cancer surgery in the tables.

As you pointed out, the tables were complicated and difficult to read, so we have made some revisions.

Reviewer 3 Report

Comments and Suggestions for Authors
  1. It is recommended to supplement the literature screening process with a PRISMA flow diagram.
  2. The text mentions that RAS has a shorter learning curve, but it does not quantify the number of cases required to master different procedures (e.g, total gastrectomy vs. distal gastrectomy). It is suggested to cite data from CUSUM analysis studies on learning curves to enhance persuasiveness.
  3. The health economics of RAS is not addressed. It is recommended to add content comparing the cost-effectiveness of robotic surgery with traditional procedures.
  4. The future development directions of RAS can be further expanded, such as exploring the potential integration of cutting-edge technologies like AI-assisted intraoperative navigation and 5G remote surgery.

Author Response

We thank the reviewer for their thoughtful and constructive feedback, which has helped us improved our manuscript. Please see the detailed responses to each of the comments and the corresponding revisions made to the manuscript.

COMMENT 1.

It is recommended to supplement the literature screening process with a PRISMA flow diagram.

Reply:  Because this article is a narrative review, we did not consider the PRISMA flow, which is required for systematic reviews or meta-analyses, to be necessary and did not supplement it. Please forgive us for the editing, which has been done with consideration given to its position as a narrative review.

COMMENT 2.

The text mentions that RAS has a shorter learning curve, but it does not quantify the number of cases required to master different procedures (e.g, total gastrectomy vs. distal gastrectomy). It is suggested to cite data from CUSUM analysis studies on learning curves to enhance persuasiveness.

Reply: We have additionally cited the following references, which shows data from a CUSUM analysis study on the learning curve of minimally invasive gastrectomy, and have made modifications to help readers understand by describing the detailed analysis results of laparoscopic and robotic-assisted gastrectomy.

“Sun et al. [121] reported that the learning curve for LG was overcome after approximately 20 cases, whereas only 10 cases were required for RG. In contrast, Kim et al. [122] found that 25 cases were required to surpass the learning curve and achieve sufficient proficiency in RG. Both studies indicated that prior LG experience and the mode of training influence the progression through the learning curve [121, 122]. Additionally, Zheng et al. [123] reported that 22 cases were required to overcome the learning curve for robotic distal gastrectomy and 20 cases for robotic total gastrectomy, suggesting that proficiency levels may vary slightly between different robotic procedures.”  (Page 13, Line 411-418)

References

  1. Sun, L, F.; Liu, K.; Su, X, S.; Wei, X.; Chen, X, L.; Zhang, W, H.; Chen, X, Z.; Yang, K.; Zhou, Z, G.; Hu, J, K. Robot-Assisted versus Laparoscopic-Assisted Gastrectomy among Gastric Cancer Patients: A Retrospective Short-Term Analysis from a Single Institution in China. Gastroenterol Res Pract. 2019, 9059176.
  2. Kim, M,S.; Kim, W, J.; Hyung, W, J.; Kim, H, I.; Han, S, U.; Kim, Y, W.; Ryu, K, W.; Park, S. Comprehensive Learning Curve of Robotic Surgery: Discovery From a Multicenter Prospective Trial of Robotic Gastrectomy. Ann Surg. 2021, 273, 949-956.
  3. Zheng-Yan, L.; Feng, Q.; Yan, S.; Ji-Peng, L.; Qing-Chuan, Z.; Bo, T.; Rui-Zi, G.; Zhi-Guo, S.; Xia, L.; Qing, F.; et al. Learning curve of robotic distal and total gastrectomy. Br J Surg. 2021, 108, 1126-1132.

COMMENT 3.

The health economics of RAS is not addressed. It is recommended to add content comparing the cost-effectiveness of robotic surgery with traditional procedures.

Reply: In the section “5.1. Limitations and barriers to RAS implementation”, we briefly mentioned the medical economic aspects of RAS by citing the paper by Sakai et al. We have added a detailed comparison of the cost-effectiveness of laparoscopic surgery and RAS as follows:

“Sakai et al. conducted a comparative cost analysis of robot-assisted versus laparoscopic surgery to evaluate the financial impact of each procedure. They found that RAS generally yielded lower profits than laparoscopic procedures, with robot-assisted proximal gastrectomy even resulting in a negative gross surgical profit. The high costs of robotic instruments and ongoing maintenance were not adequately offset by government-determined reimbursement rates, leading to financial losses in these cases. Additionally, they noted that more complex procedures, such as esophageal, rectal, bladder, and prostate surgeries, generated higher gross profits. The financial feasibility of RAS may vary depending on the healthcare system and the specific surgical procedure, with potential economic benefits emerging in the future.” (Page 14, Line 443-452)

COMMENT 4.

The future development directions of RAS can be further expanded, such as exploring the potential integration of cutting-edge technologies like AI-assisted intraoperative navigation and 5G remote surgery.

Reply: In the "5. Challenges and developments in RAS systems" section, the following citation and references have been added:

" The integration of AI and machine learning (ML) into RAS is expected to enhance intraoperative decision-making by improving the identification of fine and complex anatomical structures. These technological advancements may improve short-term surgical outcomes. However, several challenges remain, including the high cost of RAS, complexity of system maintenance, and large physical footprint of the equipment. Future innovations—such as AI-driven automation, nanorobots, microdissection techniques, semi-automated remote robotic systems, and the application of 5G connectivity for remote surgery—have the potential to further expand the clinical utility of RAS. The development of telesurgery capabilities may allow the introduction of RAS to medically underserved regions, thereby helping bridge the evidence–practice gap in cancer treatment, reduce delays in surgical skill acquisition, and mitigate the global shortage of trained surgeons [118–120].”  (Page 13, Line 393-404)

References

  1. Chatterjee, S.; Das, S.; Ganguly, K.; Mandal, D. Advancements in robotic surgery: innovations, challenges and future prospects. J Robot Surg. 2024, 18, 28.
  2. Ebihara, Y.; Hirano, S.; Kurashima, Y.; Takano, H.; Okamura, K.; Murakami, S.; Shichinohe, T.; Morohashi, H.; Oki, E.; Hakamada, K.; et al. Tele-robotic distal gastrectomy with lymph node dissection on a cadaver. Asian J Endosc Surg. 2024, 17, e13246.
  3. Takahashi, Y.; Hakamada, K.; Morohashi, H.; Akasaka, H.; Ebihara, Y.; Oki, E.; Hirano, S.; Mori, M. Reappraisal of telesurgery in the era of high-speed, high-bandwidth, secure communications: Evaluation of surgical performance in local and remote environments. Ann Gastroenterol Surg. 2023, 7, 167-174.